# UniRiT: Towards Few-Shot Non-Rigid Point Cloud Registration

## Abstract

Non-rigid point cloud registration is a critical challenge in 3D scene understanding, particularly in surgical navigation. Although existing methods achieve excellent performance when trained on large-scale, high-quality datasets, these datasets are prohibitively expensive to collect and annotate, e.g., organ data in authentic medical scenarios. With insufficient training samples and data noise, existing methods degrade significantly since non-rigid patterns are more flexible and complicated than rigid ones, and the distributions across samples are more distinct, leading to higher difficulty in representation learning with few data. In this work, we aim to deal with this challenging few-shot non-rigid point cloud registration problem. Based on the observation that complex non-rigid transformation patterns can be decomposed into rigid and small non-rigid transformations, we propose a novel and effective framework, UniRiT. UniRiT adopts a two-step registration strategy that first aligns the centroids of the source and target point clouds and then refines the registration with non-rigid transformations, thereby significantly reducing the problem complexity. To validate the performance of UniRiT on real-world datasets, we introduce a new dataset, MedMatch3D, which consists of real human organs and exhibits high variability in sample distribution. We further establish a new challenging benchmark for few-shot non-rigid registration. Extensive empirical results demonstrate that UniRiT achieves state-of-the-art performance on MedMatch3D, improving the existing best approach by 94.22%.

## 1 Introduction

Non-rigid point cloud registration(N-PCR) is a fundamental problem in 3D scene understanding, with significant applications in motion estimation (Liu et al., 2019; Shen et al., 2023), reconstruction (Newcombe et al., 2015; Das et al., 2024), robotic manipulation (Yin et al., 2021; Weng et al., 2022), and surgical navigation (Baum et al., 2021; Golse et al., 2021). In contrast to rigid registration, which confines point cloud transformations to rotation and translation (Yew & Lee, 2020; Qin et al., 2022), N-PCR demands the application of distinct displacements to individual points (Baum et al., 2021; Li & Harada, 2022), consequently producing a wide range of transformation patterns (Wang et al., 2017). This complexity makes N-PCR significantly more challenging to achieve.

Despite prior learning-based N-PCR methods have demonstrated significant potential on various general benchmarks (Li & Harada, 2022; Liu et al., 2024; Yu et al., 2023), their success typically relies on large-scale training datasets with similar distributions and noise-free conditions (Lv et al., 2018; Li et al., 2021). However, such assumptions are often unrealistic in real-world scenarios. A typical example is the acquisition of organ point clouds (Devi & Bansal, 2024). The collection of medical data requires the involvement of specialized medical personnel and involves patient privacy concerns. Moreover, organ point clouds collected from different patients often exhibit substantial variability, leading to limited-scale medical datasets with significant distributional differences. Organ point clouds are typically captured using CT and MRI technologies (Li et al., 2023), and the complex internal structures of organs, combined with the dynamic nature of the scanning process, inevitably introduce noise and result in incomplete structural capture. This exacerbates the challenge of NPCR. Although non-learning-based methods do not rely on training datasets, their computational inefficiency severely limits their practical applicability. Our experimental results indicate that existing N-PCR methods cannot adapt well to such real scenarios.

To simulate realistic N-PCR scenarios (Zhang et al., 2024), we propose a new benchmark, Med-Match3D, based on a 3D point cloud dataset of real human organs (Li et al., 2023), which comprises a total of 3,408 pairs of registered point clouds across 10 different human organ types. Our study demonstrates that existing registration methods (Li & Harada, 2022; Yu et al., 2023) perform well on individual liver datasets but show suboptimal performance on the larger MedMatch3D dataset. To explain this phenomenon, we re-examined the issue of few-shot point cloud registration (Zhao et al., 2021; Kang & Cho, 2022). Although all samples belong to the same organ type, significant distributional differences may exist between different samples. In some cases, the intra-organ variability may even exceed the inter-organ variability. This suggests that a single organ type may present a wide range of transformation patterns, which increases the difficulty of network learning. Consequently, the core challenge is no longer merely aligning different organ types, but enabling the model to generalize to a variety of complex and sparsely sampled transformation patterns under few-shot conditions. Based on this analysis, we define the problem of few-shot N-PCR.

Learning the complex transformation patterns inherent in N-PCR (Baum et al., 2021; Wu et al., 2020) presents significant challenges for neural networks, particularly in small-sample datasets with notable distributional differences, which may result in training samples that fail to cover all possible pattern variations. Consequently, test samples may exhibit significantly different transformation patterns than those encountered during training. A promising solution is to decompose these complex patterns into simpler fundamental patterns. Non-rigid pattern can often be characterized as a combination of rigid and smaller non-rigid movements. By applying a unique displacement vector to each point in the target point cloud, we can derive the source point cloud. First, we apply a rigid transformation to align the centroid of the source point cloud with that of the target point cloud, followed by non-rigid registration. This two-step approach converts the unconstrained non-rigid registration problem into a more manageable one, where small adjustments are made to individual points while keeping the centroid fixed. This significantly reduces the complexity of transformation patterns and eases the N-PCR process. Based on this framework, we propose UniRiT, a joint model for few-shot N-PCR. We utilize Gaussian Mixture Model (GMM) to analyze the registration process from the perspectives of data distribution similarity and generalization.

**Our contributions are as follows:** (1) We systematically study a new task of few-shot N-PCR for data-scarse scenarios. To the best of our knowledge, this is the first work to define and address this new task. (2) Having observed that the complex non-rigid patterns in point clouds can be decomposed into a combination of rigid pattern and non-rigid refinement, we present a two-step registration approach to simplify the learning process for complex transformation patterns. (3) We establish a new benchmark based on a real human organ 3D point cloud dataset, MedMatch3D, for few-shot N-PCR. Extensive results demonstrate that our method is simple yet effective, achieving state-of-the-art performances and achieving substantial performance improvements over existing approaches on the challenging dataset, verifying its effectiveness in few-shot and high-noise scenarios.

## 2 RELATED WORKS

### 2.1 NON-RIGID POINT CLOUD REGISTRATION METHOD

The objective of non-rigid point cloud registration is to estimate a deformation matrix that can be applied to the source point cloud to map it to the target point cloud. Coherent Point Drift (CPD) (Myronenko & Song, 2010) formulates point cloud registration as a probability density estimation problem, but it is sensitive to occlusions and outliers. Bayesian Coherent Point Drift (BCPD) (Hirose, 2020) enhances the robustness of CPD through variational inference but is prone to local minima. For learning-based methods, FPT (Baum et al., 2021) is a non-rigid point cloud registration approach for the prostate, which achieves high efficiency due to its simple architecture, but lacks robustness in complex scenarios. Lepard (Li & Harada, 2022) employs a Transformer architecture to estimate point correspondences, followed by N-ICP (Serafin & Grisetti, 2015) for registration, providing high-quality encoding but with slower processing speed. RoITr (Yu et al., 2023) introduces rotation-invariant attention into an encoder-decoder framework to improve point correspondence estimation. Scene flow estimation is a problem similar to non-rigid registration that involves predicting point-level displacements. Relevant methods include PointPWC-Net (Wu et al., 2020), which captures fine-grained motion through iterative refinement, albeit with high computational costs. In contrast,

BPF (Cheng & Ko, 2022) leverages bidirectional learning to enhance the robustness of the model when dealing with outliers and partial correspondences.

## 2.2 FEW-SHOT POINT CLOUD LEARNING

Given the complexity and labor-intensive nature of point cloud data collection, the importance of few-shot point cloud learning has become increasingly evident. Previous few-shot point clouds learning tasks have primarily focused on classification and segmentation. The pioneering work attMPTI (Zhao et al., 2021) leverages label propagation to exploit the relationship between prototypes and query points. BFG (Mao et al., 2022) introduces bidirectional feature globalization to activate the global perception of both prototypes and point cloud features, thereby enhancing context aggregation. CSSMRA (Wang et al., 2023) employs a multi-resolution attention module that utilizes the nearest and farthest points to improve context aggregation. ViewNet (Chen et al., 2023) proposes a novel projection-based backbone framework, incorporating a View Pooling mechanism to boost few-shot point cloud classification performance. Additionally, SCAT (Zhang et al., 2023) presents a stratified class-specific attention-based Transformer architecture, constructing fine-grained relationships between support and query features.

## 2.3 NON-RIGID POINT REGISTRATION BENCHMARK

Collecting a large-scale dataset for non-rigid point cloud registration is challenging. Existing non-rigid datasets (Bogo et al., 2014; Ye et al., 2012; Guo et al., 2015; Zuffi et al., 2017) are either limited in size or acquired through high-precision scanning, making them less applicable to real-world registration tasks. Synthetic datasets, widely used in dense optical flow methods (Mayer et al., 2016; Lv et al., 2018), such as Sinter (Butler et al., 2012), Monka (Mayer et al., 2016), and Lepard (Li & Harada, 2022), leverage rendered animations of deformable objects. While scene flow estimation utilizes real-world datasets like KITTI (Menze & Geiger, 2015), the motion changes between consecutive point clouds are often minor.

## 3 FEW-SHOT NON-RIGID POINT REGISTRATION

GMM is a commonly used probabilistic model to represent 3D point clouds (Qu et al., 2016; Yuan et al., 2020; Mei et al., 2023). Suppose that two point clouds are represented as $\mathbf{X} = \{\mathbf{x}_1, \ldots, \mathbf{x}_i, \ldots, \mathbf{x}_N\}$ and $\mathbf{Y} = \{\mathbf{y}_1, \ldots, \mathbf{y}_i, \ldots, \mathbf{y}_N\}$, where each $\mathbf{x}_i$ and $\mathbf{y}_i$ are points within the point clouds. Taking the point cloud $\mathbf{X}$ as an example, it can be modeled using a GMM, and its mathematical formulation is as follows:

$$\mathcal{G}(\mathbf{X}) = \sum_{k=1}^{K} \pi_k \mathcal{N}(\mathbf{x}|\boldsymbol{\mu}_k, \boldsymbol{\Sigma}_k), \tag{1}$$

$$\mathcal{N}(\mathbf{x}|\boldsymbol{\mu}_k, \boldsymbol{\Sigma}_k) = \frac{\exp\left(-\frac{1}{2}(\mathbf{x} - \boldsymbol{\mu}_k)^\top \boldsymbol{\Sigma}_k^{-1}(\mathbf{x} - \boldsymbol{\mu}_k)\right)}{(2\pi)^{d/2}|\boldsymbol{\Sigma}_k|^{1/2}}, \tag{2}$$

where $K$ is the number of Gaussian components. $\pi_k$ is the weight of the $k$-th Gaussian component, satisfying $\sum_{k=1}^{K} \pi_k = 1$ and $\pi_k \geq 0$. $\mathcal{N}(\mathbf{x}|\boldsymbol{\mu}_k, \boldsymbol{\Sigma}_k)$ is the probability density function of the $k$-th Gaussian component.

Following the previous studies (Ma et al., 2015; Liu et al., 2021), we use two distinct GMMs to represent the two point clouds $\mathbf{X}$ and $\mathbf{Y}$. Subsequently, by calculating the divergence between the GMMs, we can approximate the distributional difference between the point clouds $\mathbf{X}$ and $\mathbf{Y}$. The $\mathcal{L}_2$ divergence between these two GMMs can be computed as follows:

$$\mathcal{L}_2(\mathbf{X}, \mathbf{Y}) = \int (\mathcal{G}(\mathbf{X}) - \mathcal{G}(\mathbf{Y}))^2 \, d\mathbf{x}. \tag{3}$$

In practical applications, directly computing this integral is challenging (Hershey & Olsen, 2007). Therefore, we estimate the divergence $\mathcal{L}_{mc}$ using the Monte Carlo sampling method:

$$\mathcal{L}_{mc}(\mathbf{X}, \mathbf{Y}) = \frac{1}{N} \sum_{i=1}^{N} (\log \mathcal{G}(\mathbf{X}) - \log \mathcal{G}(\mathbf{Y})). \tag{4}$$

| GMM $\mathcal{L}_{mc}$ | liver | brain | gall bladder | sto-mach | pan-creas | spleen | kidney |
|---|---|---|---|---|---|---|---|
| liver | 0.62 | 0.98 | 1.32 | 1.62 | 1.95 | 1.21 | 1.25 |
| brain | 0.98 | 0.76 | 0.83 | 1.52 | 1.68 | 1.29 | 1.58 |
| gallbladder | 1.32 | 0.83 | 1.40 | 2.15 | 1.75 | 1.63 | 1.05 |
| stomach | 1.62 | 1.52 | 2.15 | 1.06 | 2.76 | 2.35 | 1.31 |
| pancreas | 1.95 | 1.68 | 1.75 | 2.76 | 2.93 | 3.19 | 2.47 |
| spleen | 1.21 | 1.29 | 1.63 | 2.35 | 3.19 | 2.07 | 1.85 |
| kidney | 1.25 | 1.58 | 1.05 | 1.31 | 2.47 | 1.85 | 0.82 |

Table 1: Comparison of $\mathcal{L}_{mc}$ values between different organs.

To quantitatively analyze the distributional differences of MedMatch3D, we use Eq. 4 to compute the distributional divergence between random samples of different organs in the MedMatch3D dataset (details are provided in appendix). Traditional datasets typically classify samples based on anatomical labels (e.g., organ types), assuming that samples within each category exhibit similar distributions. However, for a complex dataset like MedMatch3D, this assumption is overly simplistic. Due to two major challenges, samples in MedMatch3D exhibit highly diverse transformation patterns.

The first challenge is the significant distributional divergence. As shown in Table 1, even the same organ category, samples may exhibit substantial distributional differences due to patient-specific variations. This high intra-organ variability may result in deformation differences within the same organ that are even larger than those observed between different organs. Therefore, organ-level categorization cannot capture these complex patterns, and the model must learn and adapt to complex transformation patterns that extend beyond conventional organ-specific variations. The second challenge is the complexity of transformation patterns in the registration task itself. MedMatch3D focuses on aligning intra-operative and pre-operative point clouds to facilitate surgical navigation. Compared to existing benchmark datasets (e.g., sequential frames with minimal variations (Li et al., 2021; Menze & Geiger, 2015)), the morphological differences between intra-operative and pre-operative point clouds can be much larger. These pronounced shape variations lead to a significant increase in the complexity of the required transformation models (Zampogiannis et al., 2019).

Thus, the core challenge is no longer merely aligning different organ types, but enabling the model to generalize to various complex and sparsely sampled transformation patterns under few-shot conditions. The goal of few-shot non-rigid point cloud registration is to train a model that can adapt to unseen transformation patterns using only a limited number of samples with similar distributional characteristics, rather than merely fitting organ-level distributions. This redefinition highlights the practical challenges faced by registration models and emphasizes the need to effectively capture the diverse transformation patterns present in the MedMatch3D dataset.

## 4 METHODOLOGY

**Problem Definition:** In non-rigid registration, we are given a source point set $\mathbf{P}_{\mathcal{S}} = \{\mathbf{x}_1, \ldots, \mathbf{x}_i, \ldots, \mathbf{x}_N\}$ and a target point set $\mathbf{P}_{\mathcal{T}} = \{\mathbf{y}_1, \ldots, \mathbf{y}_j, \ldots, \mathbf{y}_N\}$, where $\mathbf{x}_i, \mathbf{y}_j \in \mathbb{R}^3$ represent the 3D coordinates of the points, and $N$ is their respective counts. The goal is to estimate deformation matrix $\mathbf{D}_{\text{pred}} = [\mathbf{d}_{\text{pred}_1}, \ldots, \mathbf{d}_{\text{pred}_N}]$ that align each point $\mathbf{x}_i \in \mathbf{P}_{\mathcal{S}}$ to its correspondence in $\mathbf{P}_{\mathcal{T}}$. The aligned point set $\hat{\mathbf{P}}_{\mathcal{S}} = \{\hat{\mathbf{x}}_1, \ldots, \hat{\mathbf{x}}_i, \ldots, \hat{\mathbf{x}}_N\}$ is formulated as:

$$\hat{\mathbf{x}}_i = \mathbf{x}_i + f(\mathbf{x}_i, \mathbf{D}_{\text{pred}}) = \mathbf{x}_i + \mathbf{d}_{\text{pred}_i} + \epsilon(\mathbf{x}_i), \qquad (5)$$

where $\epsilon(\mathbf{x}_i)$ represents a small adjustment term to enhance registration smoothness and robustness.

**The Challenge of Real-World Small-Sample Point Cloud Registration:** We introduce the non-rigid registration problem into real-world applications, using organ point clouds as a representative case. The success of existing one-stage non-rigid registration significantly relies on a massive number of training samples. However, when faced with few-shot scenarios with limited annotated data (e.g., organ point clouds), they suffer from a non-trivial gap of transformation patterns between limited training samples and real-world testing cases. To address these challenges, we first analyze the characteristics of non-rigid motion with GMM and illustrate the decomposition of non-rigid regis-

tration process. Based on our analysis, we design UniRIT, a dual-stage non-rigid architecture to achieve well-generalized non-rigid registration under the few-shot setting.

## 4.1 ANALYSIS WITH GAUSSIAN MIXTURE MODEL

Following previous work (Ma et al., 2015; Liu et al., 2021), we model the source point cloud $\mathbf{P}_{\mathcal{S}}$ and the target point cloud $\mathbf{P}_{\mathcal{T}}$ using two different GMMs denoted as $\mathcal{G}(\mathbf{P}_{\mathcal{S}})$ and $\mathcal{G}(\mathbf{P}_{\mathcal{T}})$. From the probabilistic perspective, the point cloud registration aims to align these two different GMMs, which minimizes the $\mathcal{L}_{mc}$ divergence between these two probability distributions, formulated as:

$$\mathcal{L}_{mc}(\mathbf{P}_{\mathcal{S}}, \mathbf{P}_{\mathcal{T}}) = \frac{1}{N} \sum_{i=1}^{N} (\log \mathcal{G}(\mathbf{P}_{\mathcal{S}}) - \log \mathcal{G}(\mathbf{P}_{\mathcal{T}})). \tag{6}$$

For clearer visaulization, we take $k = 1$ as an example to illustrate the non-rigid registration process between $\mathcal{G}(\mathbf{P}_{\mathcal{S}})$ and $\mathcal{G}(\mathbf{P}_{\mathcal{T}})$. As shown in Fig. 1a, the distributions of the source point cloud $\mathcal{G}(\mathbf{P}_{\mathcal{S}})$ and the target point cloud $\mathcal{G}(\mathbf{P}_{\mathcal{T}})$ exhibit a significant difference in terms of shapes and positions. Such a nontrivial discrepancy increases the complexity and variety of the point-cloud transformation patterns and is therefore challenging to mitigate without a large amount of training data. To ease the burden of learning such complex patterns from a limited number of training samples, we propose to decompose the conventional one-stage non-rigid registration process.

Aligning the source and target GMMs $\mathcal{G}(\mathbf{P}_{\mathcal{S}})$ and $\mathcal{G}(\mathbf{P}_{\mathcal{T}})$ can be decoupled into two steps, where a change in the mean results in a translation of the distribution and a change in the eigenvalues of the covariance matrix leads to changes in the shape of the distribution. The rigid rotation and translation of the point cloud result in a change in the mean of the distribution, while non-rigid transformations of the point cloud lead to changes in the eigenvalues of the covariance matrix. Given the rotation and translation matrix $\mathbf{R} \in \mathbb{R}^{3 \times 3}$, $\mathbf{t} \in \mathbb{R}^{3}$, the rigid transformation of GMM is formulated as:

$$\mathbf{R}^{*}, \mathbf{t}^{*} = \min_{\mathbf{R}, \mathbf{t}} \mathcal{L}_{mc}(\Psi_{\mathbf{R}, \mathbf{t}}(\mathbf{P}_{\mathcal{S}}), \mathbf{P}_{\mathcal{T}}) = \frac{1}{N} \sum_{i=1}^{N} (\log \mathcal{G}(\Psi_{\mathbf{R}, \mathbf{t}}(\mathbf{P}_{\mathcal{S}})) - \log \mathcal{G}(\mathbf{P}_{\mathcal{T}})), \tag{7}$$

$$\mathcal{G}(\Psi_{\mathbf{R}, \mathbf{t}}(\mathbf{P}_{\mathcal{S}})) = \Psi_{\mathbf{R}, \mathbf{t}}(\mathcal{G}(\mathbf{P}_{\mathcal{S}})) = \sum_{k=1}^{K} \pi_{k} \mathcal{N}(\mathbf{x}|\mathbf{R}\boldsymbol{\mu}_{k} + \mathbf{t}, \mathbf{R}\Sigma_{k}\mathbf{R}^{\top}), \tag{8}$$

where $\mathbf{R}^{*}, \mathbf{t}^{*}$ indicate the optimal solution to align $\mathcal{G}(\mathbf{P}_{\mathcal{S}})$ and $\mathcal{G}(\mathbf{P}_{\mathcal{T}})$ with rigid transformation as shown in Fig. 1b. Compared to the conventional one-step non-rigid registration, Eq. 7 is much easier to solve due to the additional rigid constraint. The result $\mathbf{R}^{*}, \mathbf{t}^{*}$ also poses a prior to the overall registration problem, which reduces the size of the following non-rigid transformation problem. Here we denote the source point cloud after the rigid transform as $\mathbf{P}'_{\mathcal{S}}$.

After the rigid transformation as shown in Fig.1b, the divergence between the probability distributions of the source and target point clouds has been significantly reduced. At this stage, we proceed with the non-rigid registration step, which assigns different displacements to each point in the source point cloud. According to the CPD (Myronenko & Song, 2010; Jian & Vemuri, 2010), the process of non-rigid registration can be interpreted as applying a component-specific transformation that maps the means $\boldsymbol{\mu}_{k}$ and covariances $\Sigma_{k}$ of each Gaussian component in the GMM. This transformation can be applied to the original GMM formulation and can be expressed as follows:

$$\min_{\mathbf{f}=\{\mathbf{f}_{\mu}, \mathbf{f}_{\Sigma}\}} \mathcal{L}_{mc}(\mathbf{f}(\mathbf{P}'_{\mathcal{S}}), \mathbf{P}_{\mathcal{T}}) = \frac{1}{N} \sum_{i=1}^{N} (\log \mathbf{f}(\mathcal{G}(\mathbf{P}'_{\mathcal{S}})) - \log \mathcal{G}(\mathbf{P}_{\mathcal{T}})), \tag{9}$$

$$\mathbf{f}(\mathcal{G}(\mathbf{P}'_{\mathcal{S}})) = \sum_{k=1}^{K} \pi_{k} \mathcal{N}(\mathbf{x}|\mathbf{f}_{\mu,k}(\mathbf{R}^{*}\boldsymbol{\mu}_{k} + \mathbf{t}^{*}), \mathbf{f}_{\Sigma,k}(\mathbf{R}^{*}\Sigma_{k}\mathbf{R}^{*\top})), \tag{10}$$

where $\mathbf{f}_{\mu,k}(\cdot)$ represents the mapping applied to the mean $\boldsymbol{\mu}_{k}$ of the $k$-th Gaussian component, and $\mathbf{f}_{\Sigma,k}(\Sigma_{k})$ represents the mapping applied to the covariance $\Sigma_{k}$ of the $k$-th Gaussian component.

As shown in Fig. 1c, after the non-rigid transformation, the transformed source point cloud has a distribution that is highly similar to that of the target point cloud. The non-rigid transformation slightly adjust to the shape of $\mathbf{P}'_{\mathcal{S}}$ while preserving a nearly unchanged location. Compared to previous

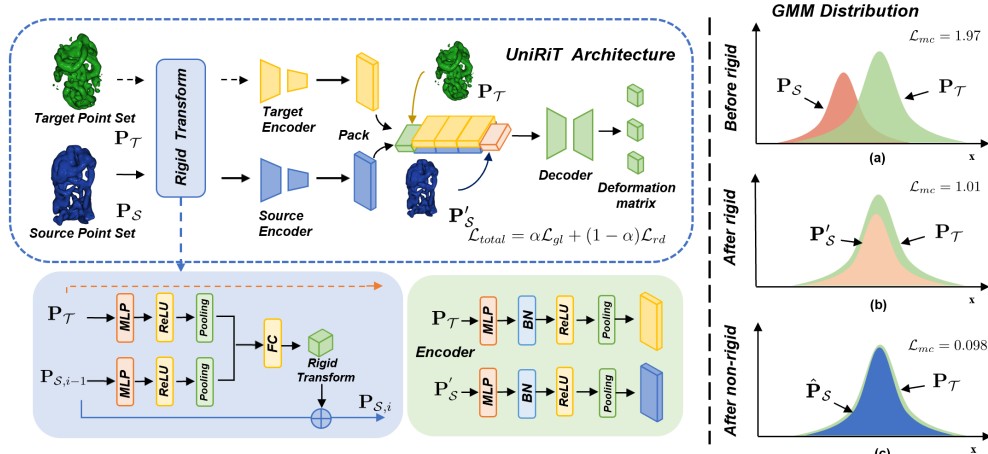

Figure 1: UniRiT performs a rigid transformation phase between the source $\mathbf{P}_{\mathcal{S}}$ and target $\mathbf{P}_{\mathcal{T}}$ point clouds, where the features of both point clouds are extracted using MLPs. These features are then passed through a decoder composed of fully connected (FC) layers, which iteratively generates rotation and translation matrices over $n$ cycles. The transformed point cloud output from the rigid module is subsequently utilized along with the target point cloud to re-extract features. These features are concatenated with the coordinate information and then input into the decoder to generate a deformation matrix, which applied to $\mathbf{P}'_{\mathcal{S}}$, yields the final transformed point cloud $\hat{\mathbf{P}}_{\mathcal{S}}$.

one-stage solutions, our decoupled alignment as in Eq. 10 regulates the non-rigid transformation primarily to local adjustments of point positions rather than the global ones and therefore reduce the complexity and difficulty of non-rigid registration problem.

## 4.2 UNIRIT ARCHITECTURE

Following our GMM-based analysis, we decompose point clouds' complex non-rigid transformation patterns into two sub-components: a unified rigid motion and a less challenging non-rigid motion. To this end, our rigid and non-rigid model forms an end-to-end foundational model for non-rigid registration, which is built on a computationally efficient architecture composed of pure MLP and fully connected (FC) layers as shown in Fig. 1. Using two MLP-based modules, the estimation of both motions can be accomplished accurately. Additionally, a rigid registration loss function is incorporated to encourage the network capture the optimal combination of the two transformation patterns, leading to a noticeable performance improvement.

In our rigid registration module, we designed a bidirectional encoding phase that enables the network to perceive the positional differences between the source point cloud $\mathbf{P}_{\mathcal{S}}$ and the target point cloud $\mathbf{P}_{\mathcal{T}}$, thereby obtaining more optimal rotation and translation matrices. MLPs with non-shared parameters are used to extract features from both the source and target point clouds. The extracted features are then concatenated and fed into FC layers to output the rigid transformation. For higher accuracy, we iterated the rigid registration process $n$ times, obtaining the output point cloud $\mathbf{P}'_{\mathcal{S}}$, where $n$ is a hyperparameter. This process can be formulated as:

$$\{\mathbf{R}_i, \mathbf{t}_i\} = \mathcal{D}_{rm}(\text{MLP}_1(\mathbf{P}_{\mathcal{S},i-1}) \oplus \text{MLP}_2(\mathbf{P}_{\mathcal{T}})), \tag{11}$$

where $\mathbf{R}_i$ and $\mathbf{t}_i$ represent the rotation matrix and translation matrix, and $\mathcal{D}_{rm}$ represents the decoder of the rigid module composed of FC layers. The index $i$ denotes the $i$-th iteration of the rigid registration process. $\oplus$ indicates the concatenation operation along the feature channel.

In the non-rigid registration stage, we adopt a bidirectional encoding scheme similar to the one used in the rigid registration stage to extract features from $\mathbf{P}'_{\mathcal{S}}$ and $\mathbf{P}_{\mathcal{T}}$, resulting in the feature representations $\mathcal{F}_{\mathbf{P}'_{\mathcal{S}}}$ and $\mathcal{F}_{\mathbf{P}_{\mathcal{T}}}$. These two features are concatenated and replicated $N$ times, where $N$ denotes the number of points in the point cloud. The original coordinates of the source and target point clouds, $\mathbf{P}'_{\mathcal{S}}$ and $\mathbf{P}_{\mathcal{T}}$, are then appended to the feature block, forming the final global feature

$\mathcal{F}^{global}$. This global feature is subsequently fed into a decoder composed of FC layers to predict the deformation matrix. The predicted deformation matrix is then applied to $\mathbf{P}'_{\mathcal{S}}$, yielding the non-rigid transformed output point cloud $\hat{\mathbf{P}}_{\mathcal{S}}$. This process can be formulated as follows:

$$\mathcal{F}^{global} = \mathbf{P}'_{\mathcal{S}} \oplus \lambda(\mathcal{F}_{\mathbf{P}'_{\mathcal{S}}} \oplus \mathcal{F}_{\mathbf{P}_{\mathcal{T}}}, n) \oplus \mathbf{P}_{\mathcal{T}}, \qquad \hat{\mathbf{P}}_{\mathcal{S}} = \mathbf{P}'_{\mathcal{S}} + \mathcal{D}_{dm}(\mathcal{F}^{global}), \tag{12}$$

where $\lambda(\rho, n)$ represents the replication of $\rho$ $n$ times and $\mathcal{D}_{dm}$ denotes the decoder of the non-rigid module composed of FC layers..

### 4.3 LOSS FUNCTION

For our rigid and non-rigid end-to-end joint architecture, we adopted a training approach that combines supervision from both the global loss function and the loss function from the rigid module.

We found that using only a global loss function may not optimally balance the allocation of rigid and non-rigid transformation. We addressed this issue by introducing a rigid loss between the output of the rigid module and the target point cloud. This ensures that the rigidly transformed point cloud aligns as closely as possible with the target point cloud during the rigid stage, thereby constraining the subsequent non-rigid transformation to a smaller range. The global loss function $\mathcal{L}_{gl}$ and the rigid module loss function $\mathcal{L}_{rd}$ can be written as:

$$\mathcal{L}_{gl} = \sqrt{\frac{1}{N} \sum_{\mathbf{x}_i \in \hat{\mathbf{P}}_{\mathcal{S}}} \min_{\mathbf{x}_j \in \mathbf{P}_{\mathcal{T}}} \|\mathbf{x}_i - \mathbf{x}_j\|^2}, \quad \mathcal{L}_{rd} = \sqrt{\frac{1}{N} \sum_{\mathbf{x}_k \in \mathbf{P}'_{\mathcal{S}}} \min_{\mathbf{x}_j \in \mathbf{P}_{\mathcal{T}}} \|\mathbf{x}_k - \mathbf{x}_j\|^2}. \tag{13}$$

During the training process, the overall loss $\mathcal{L}_{total}$ is the combination of both $\mathcal{L}_{gl}$ and $\mathcal{L}_{rd}$, balanced by a pre-defined coefficient $\alpha$ as:

$$\mathcal{L}_{total} = \alpha\mathcal{L}_{gl} + (1 - \alpha)\mathcal{L}_{rd}. \tag{14}$$

## 5 EXPERIMENTS

### 5.1 MEDMATCH3D

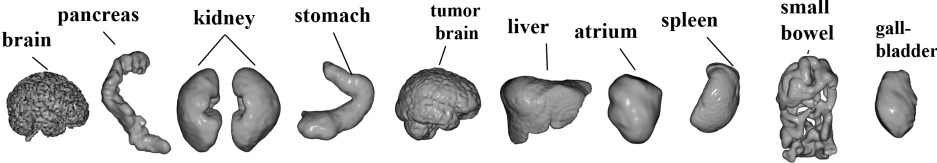

Figure 2: The ten types of the organs in MedMatch3D.

Unlike previous artificially generated or meticulously crafted high-precision datasets (Li et al., 2021; De Aguiar et al., 2008; Bogo et al., 2014), our proposed MedMatch3D dataset is derived from real human organ point clouds collected in authentic medical scenarios (Li et al., 2023). The dataset is derived from 3D depth information of organs collected using CT and MRI, which is processed and reconstructed to obtain point cloud data, making it highly representative of real-world. This approach extends the non-rigid registration problem to more realistic applications rather than focusing solely on addressing virtual shape transformation issues. We conducted a thorough review of the original 7,356 samples of 10 organ types from MedShapeNet (Li et al., 2023), uncovering a substantial number of errors and missing information in the point clouds. Specific errors can be found in the appendix. After refinement, we obtained 3,408 usable point clouds. Subsequently, we applied uniform strength TPS (Wood, 2003) deformations across all organ types, resulting in 3,408 pairs of non-rigidly registered point clouds. We conducted three sets of experiments on the Med-Match3D dataset, each with different objectives. To validate UniRiT's capability to adapt and learn from a large number of diverse samples, we performed experiments on 3,277 pairs of point clouds across nine different organs. We then employed the trained model for zero-shot testing on the small

bowel dataset, characterized by significant distribution differences, structural missingness, and real noise. This experiment aimed to assess UniRiT's robustness and superiority in registering unseen point cloud classes. Additionally, we explored the superiority of our method in handling few-shot learning problems through experiments on the small-sample liver dataset.

## 5.2 EXPERIMENTAL ANALYSIS USING THE MEDMATCH3D DATASET

We conducted experiments on the MedMatch3D dataset, covering 9 types of organs excluding the small bowel, with a total of 3,277 point cloud samples, as detailed in Table 2. As previously discussed, real-world point cloud registration tasks often involve low-quality and small-sample. The MedMatch3D dataset, collected from real human organs, aligns with these real-world constraints in terms of low quality. Many non-rigid point cloud registration methods that rely on abundant geometric information cannot be applied to our dataset (Litany et al., 2017; Donati et al., 2020), which only contains raw spatial coordinates. Given the strong alignment between our problem definition and the challenges posed by real-world data, we compared our method with scene flow estimation techniques (Cheng & Ko, 2023), as they are highly relevant to the nature of our task.

| Method | Overall Metrics | | | | RMSE (mm) | | | | | | | | |
|---|---|---|---|---|---|---|---|---|---|---|---|---|---|
| | RMSE (mm) | CD (mm) | IT (ms) | FLOPs (G) | liver | brain | kidney | tumor brain | gall bladder | sto-mach | spleen | pan-creas | atrium |
| CPD | 52.98 | 2.07 | 1063.21 | - | 3.54 | 88.18 | 45.37 | 94.94 | 22.79 | 70.25 | 53.01 | 52.12 | 37.04 |
| BCPD | 47.70 | 8.35 | 5105.63 | - | 17.75 | 80.09 | 44.23 | 85.57 | 22.49 | 69.44 | 50.28 | 55.13 | 33.51 |
| FPT | 38.98 | 20.35 | 8.23 | 7.58 | 17.14 | 66.82 | 34.11 | 71.29 | 18.79 | 54.33 | 40.10 | 42.04 | 28.78 |
| PointPWC | 39.96 | 11.35 | 31.52 | 8.91 | 19.22 | 68.38 | 34.78 | 72.68 | 19.76 | 54.64 | 40.70 | 42.62 | 30.31 |
| BPF | 38.85 | 11.92 | 33.86 | 8.16 | 14.42 | 67.25 | 34.47 | 71.49 | 20.81 | 54.63 | 40.48 | 42.55 | 29.72 |
| DifFlow3D | 42.34 | 13.35 | 90.43 | 16.97 | 30.06 | 68.43 | 35.53 | 72.94 | 20.94 | 56.10 | 41.30 | 43.72 | 30.64 |
| MSBRN | 38.32 | 12.05 | 93.02 | 18.38 | 14.17 | 67.21 | 34.41 | 71.56 | 18.63 | 54.24 | 40.07 | 41.88 | 29.42 |
| RoITr | 37.41 | 16.23 | 22.41 | 3.72 | 10.23 | 86.98 | 44.61 | 93.46 | 21.72 | 71.45 | 52.56 | 54.06 | 35.01 |
| **w/o rigid** | 8.29 | 5.01 | **7.51** | 3.19 | 12.21 | 10.22 | 6.13 | 9.01 | 8.78 | 8.54 | 6.41 | 8.19 | 7.86 |
| **UniRIT** | **2.16** | **1.88** | 18.08 | 4.58 | **2.76** | **2.74** | **1.59** | **2.82** | **1.17** | **2.98** | **1.75** | **2.21** | **1.34** |

Table 2: The performance comparison of different methods across various categories is evaluated using RMSE, chamfer distance (CD), FLOPs, and inference time (IT), where IT refers to the time required by the model to perform registration on a pair of point clouds. The best results are in **bold**, and the second best are underlined.

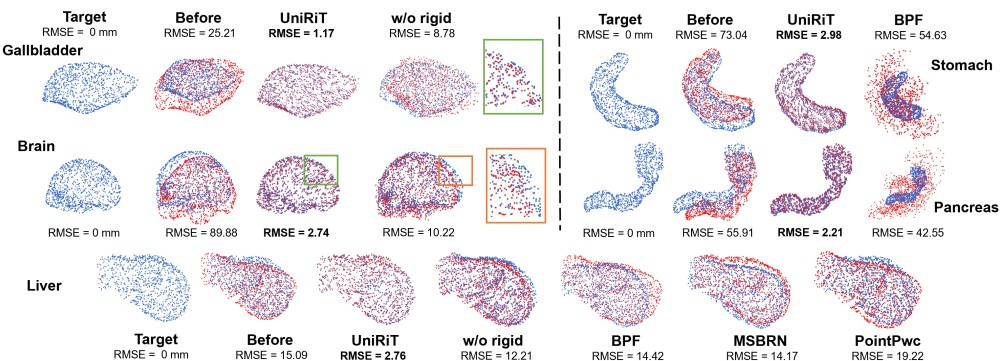

Figure 3: In the comparison of visualization results for certain organs, the differences in pre-registration RMSE across different organ types are due to their different size and complexity. The blue point cloud represents the target point cloud, while the before image illustrates the discrepancy between the source and target point clouds before registration. In the method figure, the red point cloud indicates the transformed source point cloud.

Table 2 presents the performance of UniRiT and comparative methods on the few-shot real dataset MedMatch3D. The w/o rigid refers to the version of UniRiT where the rigid module has been removed, which is used for conducting ablation studies. Most methods fail to achieve high-precision registration. Taking RMSE as an example, the registration error for FPT (Baum et al., 2021) is 38.98mm, RoITr (Yu et al., 2023) yields 37.41mm, and MSBRN (Cheng & Ko, 2023) results in

38.32mm. In contrast, UniRiT achieves a significantly lower registration error of only 2.16mm, far surpassing all existing methods and demonstrating the effectiveness of our motion decomposition strategy. During the registration process for different organs, it can be observed that the comparative methods exhibit larger registration errors, particularly for organs with complex structures or large deformation. These two factors complicate the transformation patterns of the point cloud. For instance, the registration error of MSBRN (Cheng & Ko, 2023) on the brain is 67.21 mm, which is significantly higher than the average. However, even when dealing with such complex transformation patterns, UniRiT still achieves high-precision registration with an error of only 2.74 mm.

Fig. 3 presents the registration process for some organs. Among them, BPF (Cheng & Ko, 2022) is a typical representative of comparative methods. It can be observed that the transformed point clouds become scattered. The transformation results of other scene flow estimation methods are similar to those of BPF. FPT (Baum et al., 2021) and RoiTr (Yu et al., 2023) aggregate the transformed point clouds into a dense cluster. Both phenomena indicate a failure in the registration task. We provide a detailed comparison of the remaining methods in the appendix, along with their corresponding visualizations. UniRiT is the only method that can achieve normal registration with high accuracy. Although the accuracy declines when removing the rigid module, it does not become scattered points and fails the registration. This fully demonstrates the superiority of UniRiT in the real-world challenge dataset with few samples.

## 5.3 Experimental Analysis Using a Representative Small Bowel Dataset

| Method | RMSE (mm) | CD (mm) |
|---|---|---|
| CPD (Myronenko & Song, 2010) | 109.63 | 9.38 |
| BPF (Cheng & Ko, 2022) | 99.60 | 10.51 |
| PointPWC (Wu et al., 2020) | 90.85 | 13.52 |
| RoITr (Yu et al., 2023) | 109.20 | 7.51 |
| FPT (Baum et al., 2021) | 84.45 | 36.73 |
| **w/o rigid** | 15.19 | 8.33 |
| **UniRIT** | **6.65** | **5.18** |

Table 3: Benchmark comparison on zero-shot small bowel dataset of various methods. The best results are in **bold**, while the second best are underlined.

| Method | Case A | | Case B | |
|---|---|---|---|---|
| | RMSE(mm) | CD(mm) | RMSE(mm) | CD(mm) |
| CPD | 3.54 | 2.99 | 7.65 | 4.98 |
| BCPD | 17.75 | 11.17 | 23.06 | 17.72 |
| FPT | 12.80 | 8.08 | 30.72 | 12.19 |
| PointPWC | 10.01 | 6.41 | 24.38 | 12.53 |
| BPF | 8.07 | 5.51 | 67.25 | 22.45 |
| Livermatch | 14.17 | 8.10 | 27.01 | 12.90 |
| Lepard | 8.10 | 6.02 | 12.19 | 6.98 |
| RoITr | 3.01 | 2.44 | 6.71 | 4.24 |
| **w/o rigid** | 8.29 | 7.88 | 24.82 | 12.57 |
| **UniRIT** | **2.72** | **2.31** | **3.04** | **2.83** |

Table 4: Performance comparison of different methods in Case A and Case B, evaluated by RMSE and CD. The best results are in **bold**, and the second best are underlined.

In real-world point cloud registration scenarios, challenges often arise when testing on unseen classes (Cheraghian et al., 2022). The small bowel dataset serves as a typical example for the following reasons: First, the effective sample size for the small bowel is extremely limited, consisting of only 131 samples, which is insufficient to meet the training requirements. Second, the structure and distribution of the small bowel are highly complex and diverse, coupled with the presence of external noise, resulting in low-quality samples with significant noise and substantial information loss. Fig. 4 qualitatively illustrates seven small bowel samples with significant distribution differences. Consequently, to evaluate the generalization and robustness of UniRiT, we conducted zero-shot testing on the challenging small bowel dataset.

Table 3 presents the quantitative results of the zero-shot testing conducted on the small bowel dataset, evaluated using RMSE. The FPT (Baum et al., 2021) method yielded an error of 84.45 mm, while RoITr (Yu et al., 2023) and BPF (Cheng & Ko, 2022) reported errors of 109.20 mm and 99.60 mm, respectively. The corresponding CD values for these methods were 36.73 mm, 7.51 mm, and 10.51 mm. All three methods performed poorly on the small bowel dataset and failed to complete the registration task. In contrast, UniRiT achieved a registration error of 6.65 mm, making it the only method capable of successfully performing few-shot registration on the small bowel dataset. This outcome validates the robustness and superiority of our approach.

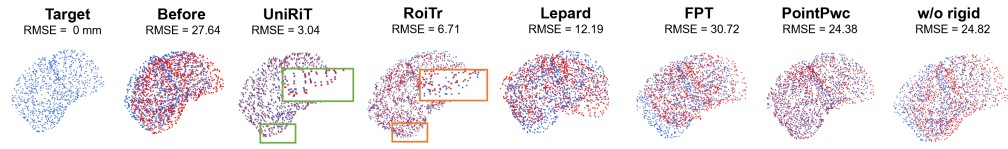

Figure 4: Seven randomly selected samples of the small bowel are shown. It can be observed that, during the acquisition of small bowel samples, issues such as incomplete structural scans and significant noise are present.

## 5.4 TESTING ON LIVER DATA WITH LARGE RIGID DISPLACEMENTS

To evaluate the performance of UniRiT on a small-sample dataset, we conducted tests on a liver dataset containing 551 samples. We utilized 487 samples as the training set, while the remaining 64 samples served as the validation and test set. Table 4 presents the corresponding quantitative results, including RMSE and CD, which are defined in the appendix. Case A in Table I describes the experimental scenario where only TPS deformation is applied, while Case B describes the experimental conditions involving significant displacement in addition to non-rigid transformation. Specifically, in Case B, random rigid rotations were applied within the range of [-45°, +45°], and displacements were randomly sampled in the normalized space of [-0.2, 0.2]. In Case A, UniRiT achieved an RMSE value of 2.72 mm, which is slightly better than RoITr's (Yu et al., 2023) RMSE of 3.01 mm and significantly outperforms other non-rigid registration algorithms. In Case B, UniRiT's RMSE value was 3.04 mm, outperforming RoITr's RMSE of 6.71 mm. This advantage may be attributed to UniRiT's combined rigid and non-rigid registration architecture, which provides a natural benefit in handling non-rigid registration scenarios involving rigid displacements.

| Target
RMSE = 0 mm | Before
RMSE = 27.64 | UniRiT
RMSE = 3.04 | RoiTr
RMSE = 6.71 | Lepard
RMSE = 12.19 | FPT
RMSE = 30.72 | PointPwc
RMSE = 24.38 | w/o rigid
RMSE = 24.82 |

Figure 5: The visualization results of Case B. For Case B, the non-rigid deformation magnitude is 15 mm, the rotation angle ranges from [0, 45°], and the translation range is [20, 30] mm.

Fig. 5 presents the registration process of UniRiT and the comparative methods in case B. RoITr, as a representative, achieved the best performance among the comparative methods with an RMSE of 6.71 mm. However, significant errors can still be observed in detailed areas, as shown in the enlarged sections of the figure. Other methods, such as Lepard (Li & Harada, 2022), PointPWC (Wu et al., 2020), and FPT (Baum et al., 2021), while not reducing to scattered points as in other organ cases in Experiment 1, still exhibit suboptimal registration results. UniRiT, on the other hand, achieves the highest registration accuracy and maintains precision in detailed regions.

## 6 CONCLUSION AND DISCUSSION

In this study, we systematically analyzed and defined the problem of few-shot non-rigid point cloud registration. Through three experimental setups—the mixed organ experiment, the zero-shot small bowel experiment, and the single small-sample liver experiment—we revealed the limitations of existing methods in handling samples with significant distributional differences, particularly in small-sample scenarios and complex transformation patterns. The introduction of the MedMatch3D benchmark dataset provides new research directions for this field and underscores the importance of considering distributional characteristics in the context of few-shot learning. Looking ahead, we plan to enrich the MedMatch3D dataset by increasing the variety and quantity of point clouds to enhance its applicability across different research challenges. Additionally, we will introduce point cloud segmentation tasks into MedMatch3D, making it a more versatile benchmark.

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

# A APPENDIX

## A.1 MEDMATCH3D

Unlike previous artificially generated or meticulously crafted high-precision datasets, our proposed MedMatch3D dataset is derived from real human organ point clouds collected in authentic medical scenarios. This approach extends the non-rigid registration problem to more realistic applications, providing a new benchmark for future methods designed for real-world non-rigid registration scenarios, rather than focusing solely on virtual shape transformation issues. We conducted a comprehensive review of the 7356 samples from 10 organ types in MedShapeNet, uncovering a substantial number of errors and missing information within the point clouds. After refinement, we obtained 3,408 usable point clouds. In the subsequent modules, we present the selected point clouds and those with errors. Subsequently, we applied uniform strength TPS deformations across all organ types, resulting in 3,408 pairs of non-rigidly registered point clouds.

**Implementation details.** All methods were implemented using the PyTorch framework on a single GPU (Nvidia GeForce RTX 4090, 24GB). The model was further fine-tuned on 1024 points randomly sampled from the original point sets, each consisting of 10,000 points. The Adam optimizer was employed, with a batch size set to 1, and the network was trained for a total of 300 epochs. For the comparative methods, we utilized their publicly available code versions and setup for epochs,optimizer and hyperparameters. To ensure fairness, all comparative methods have been thoroughly retrained on our organ datasets.

**Evaluation metrics.** To evaluate the registration quality, we use two different evaluation metrics, namely RMSE and Chamfer Distance (CD). In addition to quality evaluation, we also assess the efficiency of the model using various metrics. Inference Time (IT) refers to the time required by the trained model to process a single point cloud during testing. Furthermore, FLOPs is evaluated to measure the model's complexity and computational efficiency.

$$
\text{RMSE} = \sqrt{\frac{1}{N} \sum_{i=1}^{N} \|p_i - q_i\|_2^2}
$$

$$
\text{CD}(P,Q) = \frac{1}{|P|} \sum_{p \in P} \min_{q \in Q} \|p - q\|_2^2 + \frac{1}{|Q|} \sum_{q \in Q} \min_{p \in P} \|q - p\|_2^2
$$

(15)

## A.2 NINE TYPES OF ORGANS IN MEDMATCH3D.

As shown in Fig. 6 and Fig. 7, we present the selected samples along with the visualization results. Significant shape variations exist among different samples of the same organ; for instance, the gallbladder, stomach, and pancreas exhibit considerable distribution differences. Although the gallbladder has a relatively simple structure, its shape demonstrates the greatest diversity and distribution variability. Additionally, it is noteworthy that the point cloud structure of the liver appears relatively simple, with minimal distribution differences among various samples. This may explain the effective registration results observed with comparison methods on the liver dataset.

## A.3 FALSE SAMPLES

We identified a significant number of samples with missing information and errors in MedShapeNet. We filtered and visualized these samples, presenting the erroneous ones to provide a clearer understanding of our selection process. Fig. 8 illustrates a representative subset of the erroneous information we identified during the filtering process. The most common type of error observed is incomplete scanning, characterized by substantial gaps in the point cloud where only partial data has been captured. These incomplete point clouds exhibit significant morphological differences from normal samples, rendering them unsuitable for training and application. Consequently, we excluded these erroneous samples and retained those with more complete point cloud information.

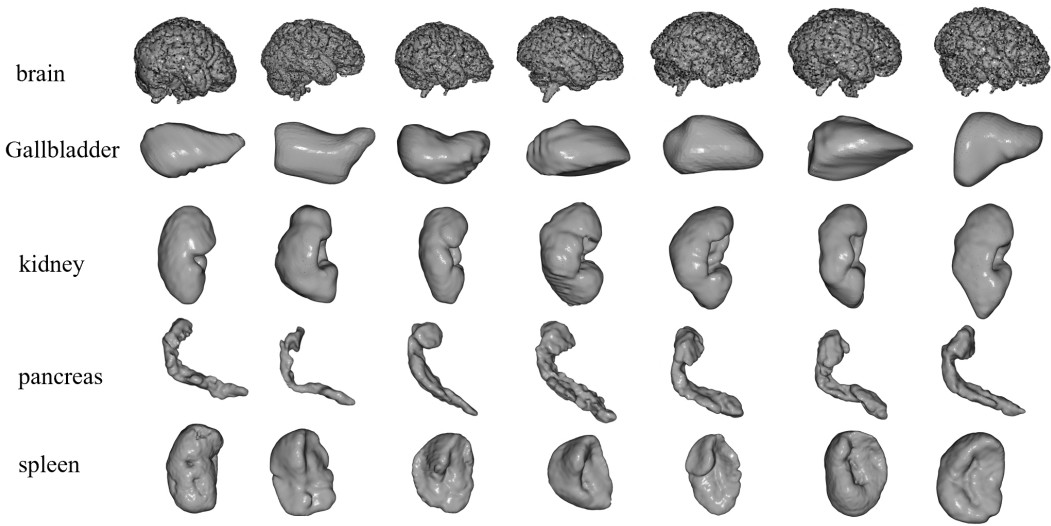

Figure 6: Visualized samples of the brain, gallbladder, kidney, pancreas, and spleen.

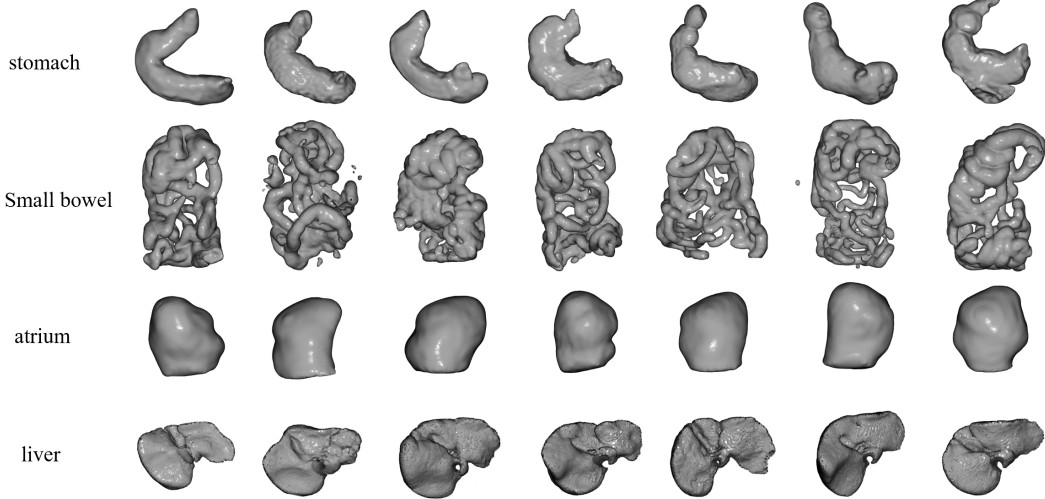

Figure 7: Visualized samples of the stomach, small bowel, atrium and liver.

### A.4 ANALYSIS OF DISTRIBUTIONAL DIFFERENCES IN ORGAN SAMPLES

To quantitatively analyze the distributional differences within organ samples and between different organ types, we randomly selected 12 groups of samples for each organ type, repeating this random selection four times. Subsequently, we utilized $\mathcal{L}_{mc}$ to calculate the differences between samples from different organs. The detailed results are shown in Table 5

### A.5 A DETAILED GRAPHICAL EXPLANATION OF GMM DECOMPOSITION OF TRANSFORMATION PATTERNS.

To better illustrate the variations of the GMM image with each module of UniRiT as shown in Figure 1, we utilized a specific pair of registered gallbladder organ images for demonstration. This figure facilitates a clearer understanding of the principles behind applying constraints to non-rigid registration as discussed in our paper.

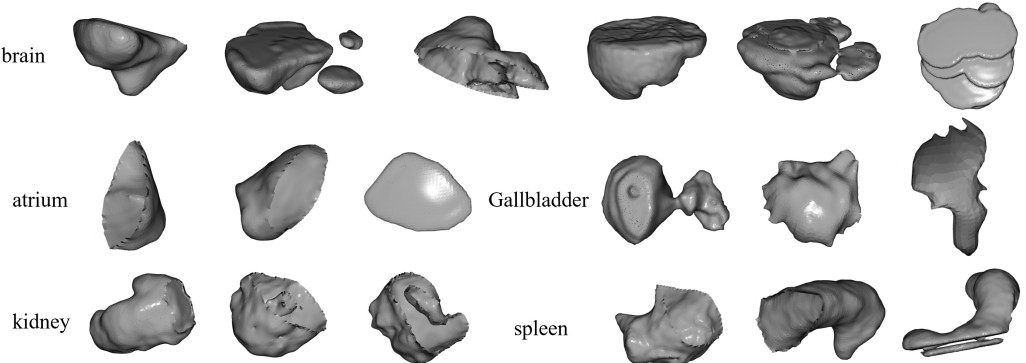

Figure 8: The visualization of erroneous and missing samples indicates severe structural loss in the point cloud, with only a minimal portion of the organ information preserved. The surface is characterized by severe perforations, leading to significant information loss.

| GMM $\mathcal{L}_{mc}$ | liver | brain | gall bladder | sto-mach | pan-creas | spleen | kidney |
|---|---|---|---|---|---|---|---|
| liver | 0.62 | 0.98 | 1.32 | 1.62 | 1.95 | 1.21 | 1.25 |
| brain | 0.98 | 0.76 | 0.83 | 1.52 | 1.68 | 1.29 | 1.58 |
| gallbladder | 1.32 | 0.83 | 1.40 | 2.15 | 1.75 | 1.63 | 1.05 |
| stomach | 1.62 | 1.52 | 2.15 | 1.06 | 2.76 | 2.35 | 1.31 |
| pancreas | 1.95 | 1.68 | 1.75 | 2.76 | 2.93 | 3.19 | 2.47 |
| spleen | 1.21 | 1.29 | 1.63 | 2.35 | 3.19 | 2.07 | 1.85 |
| kidney | 1.25 | 1.58 | 1.05 | 1.31 | 2.47 | 1.85 | 0.82 |

Table 5: Comparison of $\mathcal{L}_{mc}$ values between different organs.

The purpose of non-rigid registration is to derive a mapping transformation for the source point cloud, enabling its transformation to align with the target point cloud. The source point cloud can be modeled using GMM. The formula for GMM can be written as:

$$\mathcal{G}(\mathbf{X}) = \sum_{k=1}^{K} \pi_k \mathcal{N}(\mathbf{x}|\boldsymbol{\mu}_k, \boldsymbol{\Sigma}_k), \tag{16}$$

$$\mathcal{N}(\mathbf{x}|\boldsymbol{\mu}_k, \boldsymbol{\Sigma}_k) = \frac{\exp\left(-\frac{1}{2}(\mathbf{x} - \boldsymbol{\mu}_k)^\top \boldsymbol{\Sigma}_k^{-1}(\mathbf{x} - \boldsymbol{\mu}_k)\right)}{(2\pi)^{d/2}|\boldsymbol{\Sigma}_k|^{1/2}}, \tag{17}$$

The distribution differences between the source point cloud and the target point cloud are illustrated in Fig. 9(a), where the red and blue points represent the centroids of the source and target point clouds, respectively. The purpose of the rigid registration component is to perform an initial shape adjustment on these two point clouds to align their centroids. The GMM after the rigid transformation is given by:

$$\mathbf{R}^*, \mathbf{t}^* = \min_{\mathbf{R}, \mathbf{t}} \mathcal{L}_{mc}(\Psi_{\mathbf{R}, \mathbf{t}}(\mathbf{P}_\mathcal{S}), \mathbf{P}_\mathcal{T}) = \frac{1}{N} \sum_{i=1}^{N} \left(\log \mathcal{G}(\Psi_{\mathbf{R}, \mathbf{t}}(\mathbf{P}_\mathcal{S})) - \log \mathcal{G}(\mathbf{P}_\mathcal{T})\right), \tag{18}$$

$$\mathcal{G}(\Psi_{\mathbf{R}, \mathbf{t}}(\mathbf{P}_\mathcal{S})) = \Psi_{\mathbf{R}, \mathbf{t}}(\mathcal{G}(\mathbf{P}_\mathcal{S})) = \sum_{k=1}^{K} \pi_k \mathcal{N}(\mathbf{x}|\mathbf{R}\boldsymbol{\mu}_k + \mathbf{t}, \mathbf{R}\boldsymbol{\Sigma}_k\mathbf{R}^\top), \tag{19}$$

At this point, the transformation from Fig. 9(a) to Fig. 9(b) has been completed. The subsequent non-rigid registration process is constrained to a smaller range of motion, thereby successfully decomposing complex non-rigid transformation patterns. At this stage, only minor adjustments to

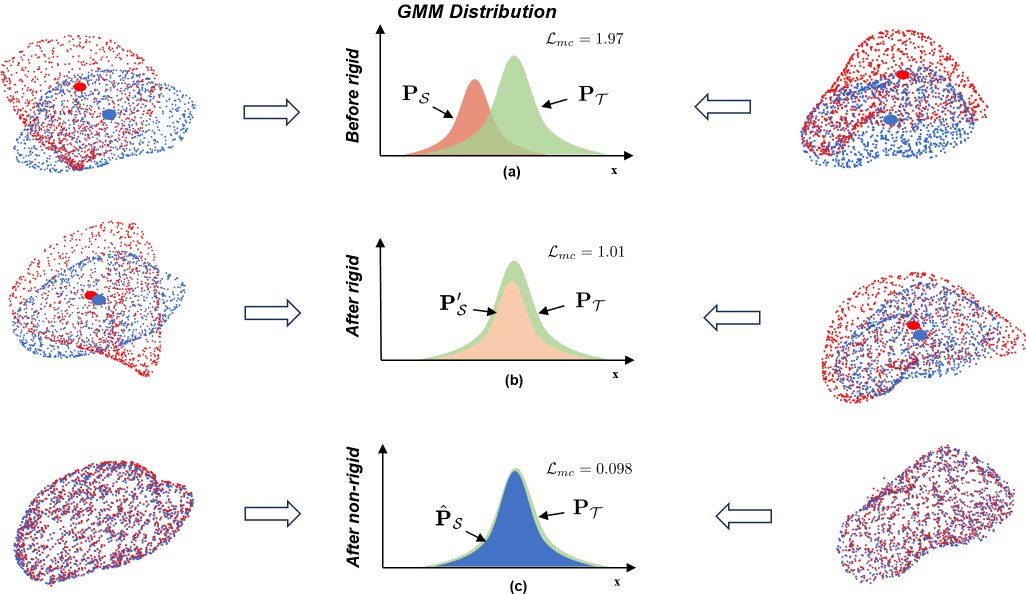

Figure 9: The GMM is employed to describe the non-rigid registration process, where the alignment is achieved by iteratively fitting a mixture model to the point clouds. In the illustration, the blue and red points denote the centroids of the source and target point clouds, respectively. The diagram presents two randomly selected test samples, positioned on the left and right, demonstrating the variation in the spatial distribution of the point clouds. By matching the centroids and minimizing the distance between them, the GMM effectively captures the transformation required for non-rigid registration.

individual points are required to achieve the transformation from Fig. 9(b) to Fig. 9(c), which can be expressed as:

$$\min_{\mathbf{f}=\{\mathbf{f}_\mu,\mathbf{f}_\Sigma\}} \mathcal{L}_{mc}(\mathbf{f}(\hat{\mathbf{P}}_\mathcal{S}), \mathbf{P}_\mathcal{T}) = \frac{1}{N}\sum_{i=1}^{N}\left(\log\mathbf{f}(\mathcal{G}(\hat{\mathbf{P}}_\mathcal{S})) - \log\mathcal{G}(\mathbf{P}_\mathcal{T})\right), \tag{20}$$

$$\mathbf{f}(\mathcal{G}(\hat{\mathbf{P}}_\mathcal{S})) = \sum_{k=1}^{K}\pi_k\mathcal{N}(\mathbf{x}|\mathbf{f}_{\mu,k}(\mathbf{R}\boldsymbol{\mu}_k + \mathbf{t}), \mathbf{f}_{\Sigma,k}(\mathbf{R}\boldsymbol{\Sigma}_k\mathbf{R}^\top)), \tag{21}$$

Where $\mathbf{f}_{\mu,k}(\boldsymbol{\mu}_k)$ represents the mapping applied to the mean $\boldsymbol{\mu}_k$ of the $k$-th Gaussian component, and $\mathbf{f}_{\Sigma,k}(\boldsymbol{\Sigma}_k)$ represents the mapping applied to the covariance $\boldsymbol{\Sigma}_k$ of the $k$-th Gaussian component.

## A.6 VISUALIZE OF THE EXPERIMENTAL ANALYSIS USING THE MEDMATCH3D DATASET

Fig. 10, Fig. 11, and Fig. 12 present the comparative visualization results of UniRiT, its ablated variants, and other competing methods across different organ types, serving as a supplement to the error metrics and partial visualization results provided in the main text. Specifically, only UniRiT achieved efficient and accurate registration. In contrast, scene flow estimation methods such as MSBRN (Cheng & Ko, 2023), PointPwc (Wu et al., 2020), and BPF (Cheng & Ko, 2022) tend to transform the source point cloud into a scattered set of points, while organ-specific registration methods like FPT (Baum et al., 2021) tend to aggregate the source point cloud into a dense cluster. Similarly, the non-rigid point cloud registration method, RoiTr (Yu et al., 2023), also tends to cluster the points. These methods essentially failed to achieve successful registration. As for the ablated version of UniRiT without rigid transformations, although it did not completely fail, its registration accuracy is too low for practical applications.

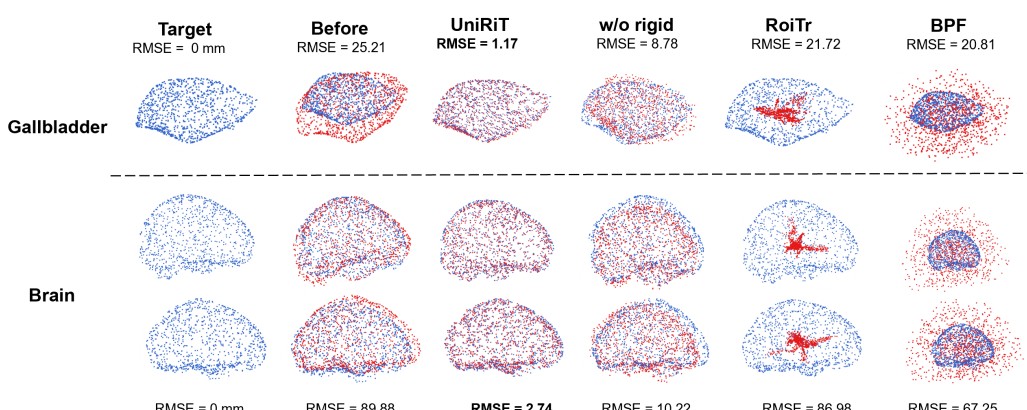

Figure 10: In the visualization comparison of certain organs, the differences in pre-registration RMSE across various organ types are attributed to variations in their size and complexity. The blue point cloud represents the target point cloud, while the before image illustrates the discrepancy between the source and target point clouds prior to registration. In the method image, the red point cloud indicates the transformed source point cloud. This figure presents a comparison against the RoiTr (Yu et al., 2023) and BPF (Cheng & Ko, 2022) methods.

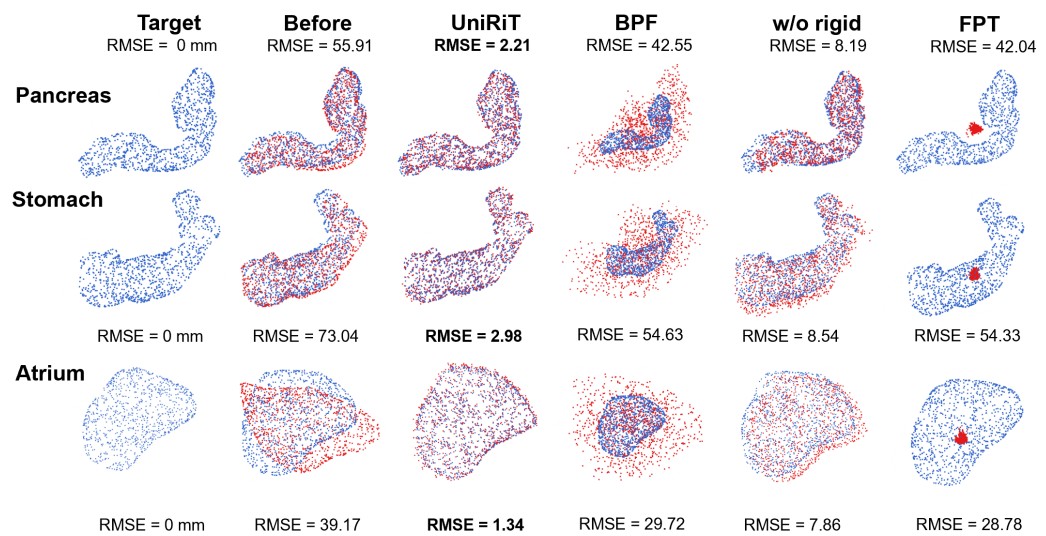

Figure 11: This figure presents the results of UniRiT and its ablation study without the rigid component, along with a comparison against the FPT (Baum et al., 2021) and BPF (Cheng & Ko, 2022) methods. The registration visualization results for the pancreas, stomach, and atrium are shown. It can be observed that BPF tends to transform the source point cloud into a scattered set of points, whereas FPT tends to aggregate them into a cluster.

### A.7 VISUALIZE OF THE ZERO-SHOT SMALL BOWEL DATASET

As previously mentioned, it is often challenging to collect a substantial amount of high-quality training datasets in the real world. A typical case is the small bowel dataset. Fig. 13 illustrates the visual samples collected during the acquisition of small bowel samples. Due to the complex structure of the small bowel and the relatively large size of the organ, it is difficult to obtain complete samples during collection; the samples are often incomplete and vary widely in their deficiencies. In this context, it is challenging to use these scarce and diverse incomplete samples for training. Therefore, conducting zero-shot testing on existing methods is of significant importance. Fig. 14 illustrates the visualization results of UniRiT alongside several comparative methods. It is evident that PointPWC,

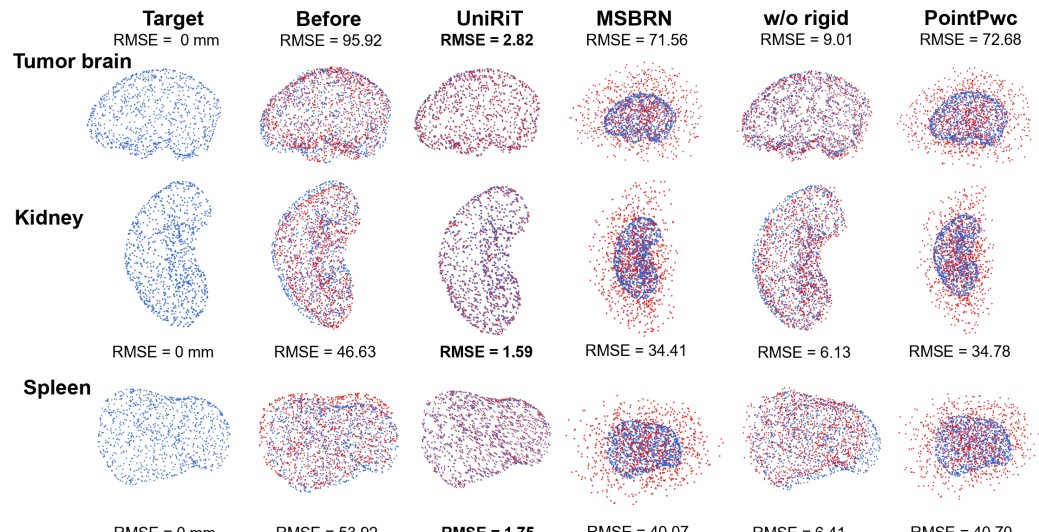

Figure 12: This figure presents UniRiT and its ablation study without the rigid component, along with a comparison against the MSBRN (Cheng & Ko, 2023) and PointPwc (Wu et al., 2020) methods. The registration visualization results for the tumor brain, kidney, and spleen are shown. This registration phenomenon further supports our previous observation that scene flow estimation methods such as MSBRN, BPF (Cheng & Ko, 2022), and PointPwc tend to transform the source point cloud into a scattered set of points, which essentially indicates a failure in the registration task.

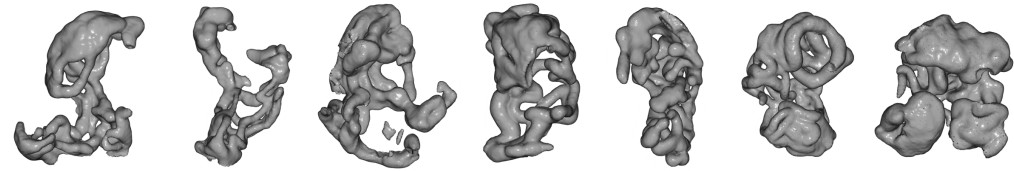

Figure 13: Seven randomly selected samples of the small bowel are shown. It can be observed that, during the acquisition of small bowel samples, issues such as incomplete structural scans and significant noise are present.

with an RMSE error of 90.85 mm, fails to perform adequately; similar scene flow estimation methods yield comparable results, transforming the point clouds into a collection of scattered points. The FPT method, with an error of 84.45 mm, tends to cluster the point clouds together, thereby losing its registration capability. Although UniRiT, when excluding the rigid module, can achieve registration, it exhibits a larger error with significant inaccuracies in finer details. In contrast, UniRiT successfully achieves accurate registration even in detailed areas.

## A.8 ABLATION EXPERIMENT

We conducted ablation experiments across three experimental groups. In the mixed experiment on the MedMatch3D dataset, the RMSE without the non-rigid module was 8.29 mm and the CD was 5.01 mm. In contrast, UniRiT achieved an RMSE of 2.16 mm and a CD of 1.88 mm, representing improvements of 73.9% and 62.5%, respectively. In the experiments on the liver dataset (Case B), the RMSE and CD improvements were 87.7% and 77.4%, respectively. These results demonstrate the superiority of UniRiT and the critical role of the rigid module, validating the effectiveness of the two-step strategy.

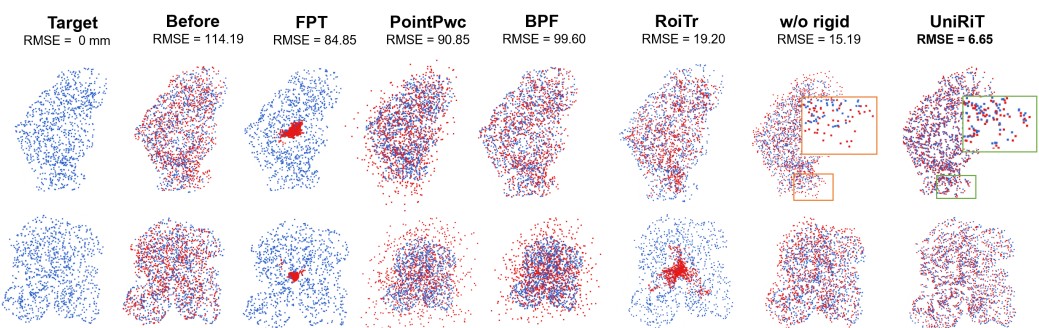

Figure 14: The results of zero-shot testing for certain methods on the small bowel dataset are presented. In the visualization, the blue point cloud represents the target point cloud. The before image shows the source point cloud before registration, while the method image displays the transformed source point cloud after applying the registration method. This comparison highlights the alignment performance and transformation effects of the respective techniques.

## A.9    ANALYSING FEW-SHOT POINT REGISTRATION

Comparing the results of Experiment 1 and Experiment 3, we found that existing methods achieved high-accuracy registration on the small-sample dataset for the single organ, liver, but failed on the mixed dataset containing multiple organs with more samples. For instance, RoITr (Yu et al., 2023) achieved an RMSE registration error of only 3.01 mm, but when trained on the comprehensive dataset, the test error for liver samples was 10.23 mm, despite the deformation of the liver being around 15 mm before registration, indicating a failure in registration. This observation seems counterintuitive. Upon re-evaluating the original concept of few-shot learning, we recognized that prior work defined few-shot samples based on human-classified organ categories, neglecting the distributional differences among samples within each organ category. In other words, samples with similar distributions map to a similar feature space, and when there is substantial distributional variation among different samples of the same organ, the network struggles to extract similarities.

Specifically regarding the liver dataset, the simple structure and small deformation of the liver result in simpler transformation patterns, making it easier for the neural network to capture such similar features. In contrast, other organs in MedMatch3D, such as the brain and pancreas, have more complex structures, while organs like the gallbladder exhibit larger deformations and distributional differences. Consequently, these organs present greater distributional discrepancies and more diverse transformation patterns. As shown in Table 1, we conducted a quantitative analysis of the distributional differences among samples within the same organ and between different organs. Our findings revealed significant distributional discrepancies among samples of the same organ, with some categories showing greater differences than those between different organs. This effectively explains the failure of existing methods in mixed datasets: although the sample size increases, the introduction of numerous samples with substantial distributional differences poses considerable challenges for the network.

