# OpenReview forum: "UniRiT: Towards Few-Shot Non-Rigid Point Cloud Registration"
_ICLR.cc/2025/Conference — ICLR 2025 Conference Withdrawn Submission_

### Official Review · Reviewer_sxHk · 2024-10-30

**Soundness:** 3
**Presentation:** 3
**Contribution:** 3
**Rating:** 5
**Confidence:** 4

**Summary:**

The paper proposes UniRiT which is a novel method for non-rigid point cloud registration. More specifically, it studies non-rigid point cloud registration in (1) data scarse, (2) few shot and (3) zero-shot scenarios, utilizing a 2-step approach (rigid + non-rigid) to simplify the learning process of the sought transformation. The evaluation of the paper is done in a new dataset (MedMatch3D) of real human 3D pointcloud of organs.

**Strengths:**

The paper presents a method to address non-rigid point cloud registration on point clouds of organs. This scenario is quite different and more challenging than deformable point-cloud registration on benchmark point clouds, e.g. [1], and also, it is not as widely studied.

Additionally, the challenge is entailed in the fact that the intra- and inter-organ deformations are way more challenging to recover and that the medical point cloud data can be scarce.

The paper also proposes and constructs a new benchmark dataset for deformable point cloud registration in a medical scenario, which might accelerate the field if made publicly available.

They test the method extensively on 3 challenging tasks using the aforementioned dataset.

[1] Li, Y., Takehara, H., Taketomi, T., Zheng, B. and Nießner, M., 2021. 4dcomplete: Non-rigid motion estimation beyond the observable surface. In *Proceedings of the IEEE/CVF International Conference on Computer Vision* (pp. 12706-12716).

**Weaknesses:**

Although I believe the paper is very interesting and attempts to address very challenging and significant tasks, I would also like to point out several weaknesses.

[A] The second contribution indicates that the paper presents a method that recovers the sought transformation in a rigid and non-rigid step, simplifying the learning process. However, I believe this is a well-studied concept in point-cloud and image registration, and many other methods first estimate a rough rigid alignment before they refine their prediction non-rigidly. I would like to invite the authors to comment on this and, if they meant something else with their second contribution, to consider revising the text since I believe it would strengthen their argument.

[B] Apart from computer vision methods that perform non-rigid point cloud registration, there are also medical point-cloud registration methods, e.g. [2] and a publicly available dataset “Lung250M-4B: a combined 3D dataset for CT-and point cloud-based intra-patient lung registration”.

I believe it would significantly strengthen the paper if the author not only cited these and other works but also experimented with the publicly available dataset to demonstrate that the proposed method is not tailored to the dataset they crafted.

Similarly, they could demonstrate results on one computer vision benchmark to prove that the method is not tailored to MedMatch3D.

[2] Heinrich, M.P., Bigalke, A., Großbröhmer, C. and Hansen, L., 2023. Chasing clouds: Differentiable volumetric rasterisation of point clouds as a highly efficient and accurate loss for large-scale deformable 3D registration. In *Proceedings of the IEEE/CVF International Conference on Computer Vision* (pp. 8026-8036).

[3] Falta, F., Großbröhmer, C., Hering, A., Bigalke, A. and Heinrich, M., 2024. Lung250M-4B: a combined 3D dataset for CT-and point cloud-based intra-patient lung registration. *Advances in Neural Information Processing Systems*, *36*.

[C] The paper constructs the MedMatch3D dataset by applying ground truth thin plate spline deformations on each organ. In other words, the deformations are artificially created by the authors and are used to supervise and assess the registration. This raises for me the following questions that I believe some of them should be enlisted as limitations of the method:

(i) How do the authors ensure the artificial deformations are realistic enough? The paper indicates that they try to address a real-world scenario, but this depends greatly on the quality of the applied artificial deformation.

(ii) Since they construct the dataset using these simulated displacements, why don’t they generate more data and limit themselves to this number?

There is always the case that I have not understood what the authors meant correctly. If this is the case, I would like to invite the authors to clarify how they constructed the dataset, what kind of data they used and what kind of deformations they tried to recover in each experiment.

[D] In the introduction the paper states “*Although non-learning-based methods do not rely on training datasets, their computational inefficiency severely limits their practical applicability.*”

I would like to invite the authors to clarify this further. I also believe that they could consider comparing their method with a non-learning-based one for completeness.

[E] The paper does not provide any code for the method or say whether the authors plan to make the code and the benchmark data publicly available. Are the authors planning to release the code and the dataset upon acceptance?

[F] 052: NPCR -> N-PCR (Typo)

**Questions:**

I would like to invite the authors to review as many points as possible from the ones I raised in the weaknesses box.
Depending on the rebuttal I am very willing to raise my score.

---

### Official Review · Reviewer_DEy6 · 2024-11-03

**Soundness:** 3
**Presentation:** 3
**Contribution:** 3
**Rating:** 5
**Confidence:** 3

**Summary:**

The authors try to address the few shot non-rigid point cloud registration problem, specifically with the applications in 3D scene understanding and surgical navigation (e.g. mapping pre-operative point clouds to inter-operative point clouds). The paper's main contributions include 1. Introduce a new benchmark dataset MedMatch3D, 2. define the problem of few-shot non-rigid point cloud registration problem, as this is often the case in the surgical navigation scenario 3. propose a two-step registration framework UniRiT.

**Strengths:**

The main contribution of the paper is introducing a new challenging problem for the researchers to look into this problem. By creating a new benchmark dataset, it will allow easier performance evaluation and comparison for the researchers working on this problem.

**Weaknesses:**

Although it is always beneficial to have new datasets, this dataset is primarily derived from a subset of an existing MedShapeNet dataset, followed by some cleaning and the application of Thin Plate Splines (TPS) to create pairs for registration. It does not add significant value to the original dataset, as the most valuable component—the real CT or MR human anatomical data—originates from MedShapeNet.

I would appreciate more visuals of the source and target organs point clouds, but based on the few examples provided in Figures 3 and 5, the deformations are minimal and not realistic for most clinical use cases.

The UniRiT framework does not introduce significant advancements to the field. The two-step strategy of decomposing non-rigid registration into rigid alignment followed by finer non-rigid adjustments is a common approach in non-rigid point clouds registration with incremental improvements. The technical contribution is not significant.

The transformations generated from the dataset hardly represent real-world deformations of organs, which might deform due to patient positioning, surgical interventions, inflation, or other factors. In the case of the small bowel, the variability of shapes can differ based on the time elapsed after food intake as well. Another common surgical scenario is data incompleteness. For example, one point cloud may have the complete liver while the other does not. Figure 4 provides some visual on this aspect but lacks of more thorough discussion.

In surgical navigation, computational overhead is also a crucial factor. The paper could include a discussion and evaluation about the computational efficiency of the proposed method.

In summary, the original MedShapeNet provides organ shapes that originate from human organs via CT or MRI. The proposed new dataset is intended for registration, but the deformations in this dataset are artificially generated. The primary contribution seems to be the "cleaning" of the data, which significantly reduces the paper's value. The dataset may not serve as a standard benchmark indicating whether an algorithm can perform well in a real surgical navigation scenario.

**Questions:**

What are the TPS parameters to generate the pair dataset?

---

### Official Review · Reviewer_XekY · 2024-11-03

**Soundness:** 2
**Presentation:** 2
**Contribution:** 3
**Rating:** 3
**Confidence:** 4

**Summary:**

The paper introduces UniRiT, a framework for few-shot non-rigid point cloud registration (N-PCR), focusing on the critical challenge of registering point clouds in data-scarce scenarios such as surgical navigation. It proposes a two-step registration strategy—initial rigid alignment followed by non-rigid refinement. The method is validated using a newly created benchmark, MedMatch3D, based on real human organ data, showing significant improvements in registration accuracy over existing methods.

**Strengths:**

- UniRiT's two-step decomposition simplifies the complex problem of N-PCR, improving the feasibility of few-shot learning.
- Demonstrated substantial performance gains, compared to optimization-based and learning-based N-PCR methods.
- The introduction of MedMatch3D provides a valuable benchmark for future research in medical applications of N-PCR.
- The paper demonstrates that UniRiT achieves competitive registration accuracy while maintaining lower computational complexity and faster runtime compared to other learning-based and optimization-based methods, making it practical for real-world applications.

**Weaknesses:**

- The two-stage registration method—rigid followed by non-rigid—is a well-established practice in image and shape registration, particularly in medical applications. This limits the novelty of UniRiT’s methodological framework.
- The framework assumes that the source and target point clouds have the same number of points, which may not hold in real-world applications. It is unclear why this restriction is necessary, especially since real-world point clouds often differ in size unless resampled. The paper does not address how the framework could adapt to varying point counts or justify the need for this assumption.
- There appear to be issues with Equations 3 and 4. In Equation 3, the integration is over $x$, which represents a 3D point, but the Gaussians are defined based on source and target points, making the formulation unclear. This raises doubts about whether it correctly represents the $L_2$ divergence between two Gaussian Mixture Models. Additionally, the discretization in Equation 4 involves a summation over index $i$, seemingly over points in the point clouds, yet the logarithms of the Gaussians are not evaluated at these points. The correctness of Equation 4 is questionable.
- Equations 3 and 4, which underpin the analysis and motivation of the proposed method, appear to be problematic. Equation 3 integrates over $x$, representing a 3D point, but the Gaussians are defined based on source and target points, leading to ambiguity in the $L_2$ divergence formulation between Gaussian Mixture Models (GMMs). This raises serious concerns about the mathematical validity of the analysis. Moreover, the discretization in Equation 4 sums over an index $i$, seemingly corresponding to points in the point clouds, but the logarithms of the Gaussians are not evaluated at these specific points. There is also a significant discrepancy from the standard  $L_2$ divergence formulation, which involves the square of the difference between densities. If this formulation is incorrect, it undermines the foundation of the entire method's motivation and results, casting doubt on the reliability of the proposed framework.
- The MedMatch3D dataset relies on simulated deformations generated using thin plate splines (TPS). However, it is unclear how challenging these deformations are and whether they realistically reflect the complexity of real-world non-rigid transformations. Additionally, there is no clear explanation of how these deformations are generated or whether they preserve the population-level variability inherent in medical data. This raises questions about the dataset's ability to truly test the robustness and generalizability of the proposed method.
- The paper begins by modeling point clouds using Gaussian Mixture Models (GMMs), suggesting that this probabilistic framework is central to the approach. However, when defining the losses, the method directly operates on the point clouds without leveraging the GMM framework. This creates confusion about the role and necessity of GMMs in formulating and designing the proposed model. The disconnect raises questions about the relevance and contribution of the GMM-based argument to the overall methodology.

**Questions:**

- Could you clarify the formulation of Equation 4? $L_2$ divergence typically involves the square of the difference between densities, not the logarithm. Why is the logarithm used here, and how does this align with the standard $L_2$ divergence? Additionally, the summation in Equation 4 does not specify where $G(X)$ and $G(Y)$ are evaluated. Shouldn't the correct Monte Carlo estimation involve sampling points and evaluating the densities at those points, as in:

$$
L_{mc}(X, Y) = \frac{1}{N} \sum_{i=1}^{N} \left( G(\mathbf{x}_i) - G(\mathbf{y}_i) \right)^2
$$

where $X_i$ and $Y_i$ are sampled points? Could you explain this discrepancy?
- How were the thin plate spline deformations in MedMatch3D designed, and what criteria were used to ensure they are sufficiently challenging?
- Do these simulated deformations capture the full range of variability found in real-world medical data, especially at the population level?
- How do you validate that the TPS-generated deformations reflect realistic clinical scenarios, and how might this affect the generalizability of your method?
- What is the specific contribution of the GMM framework to the design and functionality of the proposed model?
- Why are the losses defined directly on point clouds instead of using the GMM-based formulation introduced earlier in the paper?
- Could the model and its results have been achieved without the GMM framework, and if not, how does it enhance the performance or understanding of the method?
- How is permutation invariance specifically achieved in your use of MLPs for point cloud feature extraction? The details in the paper are not sufficient to understand how the model handles the unordered nature of point clouds effectively. Could you provide more insight into this aspect of your framework?
- Could you justify the choice of using non-shared parameters for extracting features from the source and target point clouds? How does this design decision impact the model's performance, and why was it preferred over shared parameters?

---

### Official Review · Reviewer_TZdD · 2024-11-04

**Soundness:** 2
**Presentation:** 1
**Contribution:** 2
**Rating:** 1
**Confidence:** 5

**Summary:**

This work focuses on solving the point cloud matching/registration problem with a two-step registration strategy from global rigid alignment to local non-rigid transformation.

**Strengths:**

1. Propose a straight-forward two-step shape registration method to match the point cloud from global alignment to local matching using learned transformation.
2. Provide a dataset for 3D organ shapes

**Weaknesses:**

1. The authors spent much space on explaining the gaussian mixture model and tried to related the point cloud registration with GMM as in previous works. However, in the later methodology part, the introduced network structure and the loss function has no direct relation to the GMM model. The network structure shares similarity with the point cloud segmentation models, while the loss function is similar to chamfer distance. Please explicitly explain how the GMM analysis informed their network design choices or loss function.
2. The authors proposed a straight-forward method to do the point cloud registration, combining the global right transformation and local deformation. However, the writing and structure of the paper make the reader confused. A refactor of the paper structure to make the work more straightforward would be suggested.
3. The work overly claim their contribution in few-shot definition. Their is no specific design for the few-shot learning but the testing on unseen dataset. This is more about testing the generalization ability instead of few-shot. The few-shot concept comes many time all over the paper. Please provide a clear definition of few-shot learning in the context of the work and explain how the method specifically addresses few-shot scenarios.
4. With the simple network design (similar to the pointnet), it is hard to imagine such a improvement over the existing methods (94.22% claimed). In the manuscript, there is no clear explanation about the training setting for the existing model. Please provide the training settings for both their method and the baselines
4.1. Whether baseline models were retrained on the same data or if pre-trained weights were used
4.2. A discussion of potential factors contributing to the large performance gap
4.3. Statistical significance tests for the reported improvements

5. I need to see more experiments. 1) how the proposed method performs on the existing non-medical datasets. 2) ablation study about the improvement brought by the global rigid transformation and local transformation. For example,
5.1. Non-medical datasets that would be relevant for comparison (e.g., ModelNet40, ShapeNet)
5.2. Ablation study design, such as comparing performance with and without the global rigid transformation step
5.3. Asking for an analysis of the method's performance on pre-aligned shapes compared to SOTA methods, which could isolate the contribution of the local transformation component.

**Questions:**

Solve the questions on the weakness part.

---

### Note · Authors · 2024-11-13

I have read and agree with the venue's withdrawal policy on behalf of myself and my co-authors.